

Propagating characteristics of mesospheric gravity waves observed by an OI 557.7 nm
airglow all-sky camera at Mt. Bohyun (36.2°N, 128.9°E)
Jun-Young Hwang[1], Young-Sook Lee[1], Yong Ha Kim[1], Hosik Kam[2], Young-Sil Kwak[2,3],
Tae-Yong Yang[2]
[1]Department of Astronomy and Space Science, Chungnam National University, Daejeon,
South Korea.
[2]Korea Astronomy and Space Science Institute, Daejeon, South Korea.
[3]Department of Astronomy and Space Science, University of Science and Technology,
Daejeon, South Korea.
Corresponding author: Young-Sook Lee yslee0923@cnu.ac.kr
Abstract
We analyzed all-sky camera images observed at Mt. Bohyun observatory (36.2°N,
128.9°E) for the period of 2017 - 2019. The image data were acquired with a narrow
band filter centered at 557.7 nm for the OI airglow emission at ~96 km altitude. The total
of 150 wave events were identified in the images of 144 clear nights. The interquartile
ranges of wavelength, phase speed, and periods of the identified waves are 20.5 - 35.5
km, 27.4 - 45.0 m/s and 10.8 -13.7 min with the median values of 27.8 km, 36.3 m/s and
11.7 min, respectively. The summer and spring bias of propagation directions of
northeast- and northward, respectively, can be interpreted as the effect of filtering by the
prevailing winds in the lower atmosphere. In winter the subdominant northwestward
waves may be observed due to nullified filtering effect by small northward background
wind or secondary waves generated in the upper atmosphere. Intrinsic phase speeds and
periods of the waves were also derived by using the wind data simultaneously observed
by a nearly co-located meteor radar. The nature of vertical propagation was evaluated in
each season. The majority of observed waves are found to be freely propagating, and
thus can be attributed to wave sources in the lower atmosphere.



Keywords: atmospheric gravity wave, horizontal and vertical propagation, Doppler effect,
filtering effect, wave ducting

1.Introduction
Short-period atmospheric gravity waves (<100 min) are well known for playing an important
role in carrying energy and momentum from the lower atmosphere, upward propagating and
depositing them into the mesosphere and lower thermosphere (MLT) region (Lindzen, 1981;
Fritts and Alexander, 2003). In the mid- and high-latitude MLT region the transported energy
and momentum are deposited through the breaking and dissipating processes of gravity waves,
and affect significantly the zonal flow in both hemispheres, which in turn causes the pole-to-
pole circulation resulting in the cold and warm mesosphere in summer and winter, respectively
(Lindzen, 1981; Fritts and Vincent, 1987; Fritts and Alexander, 2003; Becker, 2012).
Atmospheric gravity waves are generated by a number of causes or mechanisms, such as
mountainous terrain, convective activity triggered by severe weather phenomena, wind shear,
and areas with high baroclinic instability (Fritts and Alexander, 2003).
The characteristics of short-period gravity waves have been studied by the observation of
airglow emission in the MLT region. Airglow imaging technique has been developed to observe
gravity waves directly by using a wide-field or all-sky lens with a highly sensitive cooled
charge-coupled device (CCD) detector. The observation using an all-sky camera has an
advantage of being able to derive various parameters of gravity waves through series of
processing in time and spatial domains. Time series of all-sky airglow images can be converted
into series of 2-dimensional image arrays that can be analyzed objectively to obtain the
horizontal wavelength, propagation phase speed and period of the wave (Taylor et al., 1993).
An all-sky imager had been deployed at Bohyun observatory (BHO, 36.2°N, 128.9° E) to
observe various airglows, including OH Meinel 720-910 nm, $O_2$ atmospheric band near 865.7
nm, OI 630 nm and OI 557.7 nm in the pilot period of 2002 - 2005. Later, the all-sky camera
at BHO focused on the OI 557.7 airglow observation because the throughput of the OI 557.7
filter is far more efficient than other filters. The previous studies with the all-sky observation
at BHO have reported seasonal variation of wave parameters and horizontal propagation
directions (Kim et al., 2010; Yang et al., 2015).



The characteristics of vertical propagation of gravity waves can be determined by the
relationship between the horizontal phase speed of gravity waves and the background wind
field, and vertical temperature profile. The nature of vertical propagation can be classified into
critical-level filtering, ducting, and freely propagating modes. Critical-level filtering effect is
caused when the horizontally propagating wave meets with the same vector of background
wind, and the wave would be absorbed or reflected out (Kim and Chun, 2010; Heale and
Snively, 2015). The wave that is reflected from the upper and (or) lower altitude regions can
be (partially) ducted (Fritts and Alexander, 2003). The wave ducting can occur when the wave
propagates against background wind field, at which background wind profile has a local
maximum, called Doppler ducting (Chimonas and Hines, 1986; Isler, 1997; Nappo, 2002;
Suzuki et al., 2013). In addition, large vertical changes of background winds such as wind shear
or curvature wind can provide a favorable condition to cause Doppler ducting (e.g., Chimonas
and Hines,1986; Isler et al., 1997). The ducted wave can horizontally propagate much longer
distance than freely propagating waves (Isler et al., 1997; Hecht et al., 2001, 2004; Pautet et
al., 2005). In freely propagating mode, horizontally propagating waves can be Doppler shifted
by opposing or forwarding background wind. Therefore, the background wind can play a
crucial role in evaluating the nature of both vertical and horizontal propagation of gravity
waves. Fortunately, we were able to take advantage of the background wind measurements
around the OI airglow layer by a meteor radar at Gyeryong nearby BHO.
This study reports the characteristics of the apparent and intrinsic parameters of observed
gravity waves by using the all-sky imaging data for the period of 2017 – 2019 along with the
mesospheric wind data that were simultaneously observed by a meteor radar. The intrinsic
parameters of gravity waves allow to understand the relation between the observed wave
directions and the background winds as well as the nature of vertical propagation at mid-
latitude mesosphere around the east Asia.
2. Observational and model data
We analyze OI 557.7 airglow images observed by the all-sky camera at BHO from April
2017 through December 2019. Images of the total 144 nights were used in the analysis by
excluding the cases of cloudy and moon-lit nights and equipment malfunction. The all-sky



camera at BHO is an ultra-high speed (f/0.95) 3-inch camera composed of a fisheye lens with
a viewing angle of 180°, telecentric lens to adjust airglow emission light path for parallel
incident to filter, a 6-position filter wheel installed with two narrowband filters (OI-557.7, OI-
630.0), and a 1024×1025 CCD detector. The detail description of the all-sky camera at BHO
has been given in Yang et al. (2015). The images with the OI 557.7 filter were obtained
continuously at intervals of 5 minute with an exposure time of 90-150 seconds and a spatial
resolution of 500 km radial region. The OI-630.0 nm filter was not used in this period of
observation.
A very-high frequency (VHF) meteor radar system has been operating at Gyeryong station
(36.2˚N, 127.1˚E), since November, 2017. The Enhanced Meteor Detection Radar (EMDR)
system (supplied by ATRAD Pty Ltd) is an interferometric radar consisting of five channels.
The system is operated with specifications of a transmitter peak power of 24 kW, duty cycle of
8.4 % from 2017/11-2018/05 and 4.2 % from 2018/05 to the present. The meteor radar provides
meridional and zonal winds at 2 km bin in the 80 – 100 km altitude range every hour. The wind
data were utilized when the intrinsic wave parameters and vertical propagation were examined.
In addition, the temperature information between 80 and 100 km was adopted from the
NRLMSIS2.0 model when the Brunt- Väisälä frequency was computed to evaluate the vertical
wavelengths of gravity waves.

3. Data processing for acquiring wave parameters

112       The procedure to acquire the wave parameters can be separated into two steps: pre-

processing of all-sky camera images and the image processing with 2D image. The pre-
processing includes image selection of clear nights (see Figure 1a), star removing, transforming
fisheye lens image into the horizontal plane image (500 km×500 km, see Figure 1b) at the OI
557.7 airglow altitude of 96 km. The details of pre-processing method are provided in Kam

117   (2016).

Time series of pre-processed images were first converted into time-difference images (Figure
2a), from which large-scale modulation was removed by applying 2D bandpass filtering
(Figure 2b). The time-difference (TD) image is obtained from two consecutive images (see
Figure 1b) by subtracting from one to another. We then applied 2D Fast Fourier Transformation



(FFT) to derive wave parameters of quasi-monochromatic waves from the series of TD images
(Tang et al, 2005).
The 2D FFT operation of two TD images produces 2D spectrum arrays of J1 and J2, which can
be cross-correlated as in Equation 1 to derive a phase difference $(\phi_1 - \phi_2)$ of the wave

$$f(k_x, k_y) = J_1(k_x, k_y)J_2^*(k_x, k_y) = R_1 R_2 \exp(i\,(\phi_1 - \phi_2\,)), \qquad (1)$$

where $k_x$ and $k_y$ are zonal and meridional wave numbers, respectively. The value of
$|R_1 R_2|^2$ derived from $\left|f(k_x, k_y)\right|^2$ represents the magnitude of the wave. The dominant wave
was chosen at the maximum magnitude, whose $k_x$ and $k_y$ provide the wavelengths of the
dominant wave. Along with the time difference and the wavelength information, the phase
difference allows to determine the observed phase speed of the dominant wave.

4. Characteristics of observed waves at Mt. Bohyun
The total of 150 wave events were identified from the all-sky image data for 3 years (2017-
2019). For these wave events, horizontal wavelength, observed phase speed, observed period,
and propagation direction of the dominant wave are derived and their distributions are plotted
as in Figure 3. The interquartile range (IQR) of wavelength is spanned from 20.5-35.5 km with
a median value of 27.8 km; the observed phase speed IQR is from 27.4-45.0 m/s with a median
of 36.3 m/s, and the period IQR is from 10.8-13.7 with a median of 11.7 min. In addition, the
predominant propagating directions are north (44%) and northeast (33%). The characteristics
of these wave parameters were similar to the results of Kim et al. (2010).
In order to compare consistently the results of Takeo et al. (2017), which reported the similar
observation in the east Asia, we divided seasons in the same way: from February 21 - April 19
(2 months) for spring, from April 20 - August 20 (4 months) for summer, from August 21-
October 20 (2 months) for fall, and from October 21 - February 20 (4 months) for winter.
Seasonal wave propagation vectors and their occurrences are shown in Figures 4a-d. The
seasonal occurrences for observed (yellow) horizontal wavelength, observed phase speed, and
observed period, and intrinsic (green) phase speed and period are shown in Figure 5. The





median values of the parameters for each season are summarized in Table 1. In spring, the
propagation primarily to the northeast and next the north takes up 35% and 24% out of 29 cases,
respectively, as shown in Figure 4a. In summer, propagation directions to the north (50%) and
northeast (35%) are dominant (Figure 4b). In fall, the wave seems to propagate all-direction
without preference (Figure 4c). The fall season contains particularly small number of wave
events due to equipment problem and poor weather. In winter, the propagation directions seem
to be grouped into the south (27%), northwestward (23%) and southwest (16%) (see Figure
4d). In terms of the median values, the observed phase speed in winter is particularly slower
than other seasons, whereas other parameters show little variation. Overall it is evident that in
spring/summer, the northward and northeastward propagating gravity waves are dominant,
whereas in winter the southward and northwestward propagations are dominant. The distinct
seasonal properties of propagation direction can be attributed to the filtering effect by the
background wind field during the gravity wave propagation from the lower atmosphere (e.g.
Kim et al., 2010; Kim and Chun, 2010; Heale and Snively, 2015).
In order to confirm the filtering effect on the seasonal variation of observed propagation
direction, we checked the horizontal winds of the MERRA, version 2 (MERRA-2): MERRA-
2 is an atmospheric reanalysis model created by NASA's Global Modeling and Assimilation
Office (GMAO, https://gmao.gsfc.nasa.gov/reanalysis/MERRA-2/data_access/). MERRA-2
reanalysis data are available for 0-80 km altitudes and 0.5°×0.625° latitude and longitude
resolutions. As well known, in the spring/summer the westward wind is dominant in the middle
atmosphere, whereas in the fall/winter the eastward wind is dominant (not shown). In addition,
MR-observed annual variations of zonal and meridional winds for years of 2017-2020 are
available for 80-100 km (Kam et al., 2021). Here, prevailing winds in spring and summer are
observed in westward and southward at 80-100 km, seemingly continued from 10-80 km
altitudes, while in winter eastward winds are maintained in 80-100 km, but small northward
winds (<~10 m/s) less than 90 km turn to the southward above 90 km. It is reasonable to suggest
that westward waves in spring and summer may have been filtered out by the westward wind,
and thus are hardly observed. The southward wind in spring/summer may also have filtered out
the southward waves, which is consistent with our observation. Furthermore, in summer it is
well known that the convective system of typhoons or tropical cyclones can be significant





sources of gravity waves in the middle latitude. The typhoon-generated gravity waves in the
south of the Korean peninsula can propagate any directions, but the westward propagating
waves might be filtered out in the stratosphere by the prevalent westward wind. Therefore,
northward or northeastward propagating waves are obviously observed in Korea. The details
about typhoon-generated gravity waves can be referred to Kim and Chun (2010). In winter, it
is expected that eastward/northward waves be well filtered out by prevailing the
eastward/northward winds. However, although our observation shows southward/westward
preferential directions (see Figure 4d), northwestward waves are also subdominant. The
significant northward component of the wave direction may not be blocked by filtering effect.
In the meanwhile, it seems to survive on upward propagation up to 96 km due to the small
velocity (<10 m/s) of northward mean field. Otherwise, the northwestward wave in winter may
be interpreted as secondary waves or waves generated in the upper mesosphere.
The previous studies for mid-latitude gravity waves have reported in the majority the
dominance of eastward and northward propagations during summer (Taylor et al., 1993;
Nakamura et al., 1999; Walterscheid et al., 1999; Hecht et al., 2001; Ejiri et al., 2003; Tang et
al., 2005). Observations at BHO have confirmed the similar tendency of propagation in summer
(Kim et al, 2010; Yang et al, 2015). The summer bias of wave propagation can be distinctly
due to the critical level filtering by the prevailing zonal and meridional winds in the lower
atmosphere. However, the tendency of wave propagation also likely shows different patterns
according to localized sources. For example, in spring for Shigaraki (34.9°N, 136.1°E) Takeo
et al. (2017) observed using the OI 557.7 nm filter the dominant southwestward propagation in
addition to the northeastward that is similar to our results in Figure 4a. In winter, the southward
(equatorward) propagation was dominant in several studies although less than in summer
(Hecht et al., 2001; Ejiri et al., 2003; Tang et al., 2005). Both Ejiri et al. (2003) and Takeo et al.
(2017) observed southward dominant propagation for Shigaraki in winter. Besides, Ejiri et al.
(2003) found that winter preferential propagation may vary with latitudes because both
southward and poleward dominant propagations in both OH and OI observations were
observed at Rikubetsu (43.6°N), a relatively high latitude site.



5. Characteristics of intrinsic gravity wave parameters
The OI 557.7 nm airglow layer has been reported to be peaked at 96 km with a thickness of
~7-9 km, including both disturbed and undisturbed conditions (Vargas et al., 2007). The waves
with a vertical wavelength less than the airglow layer thickness may not be detected by an
airglow imager due to sinusoidal cancellation (Nielsen et al., 2012; Vargas et al., 2007). The
vertical wavelength of the observed wave can be derived from the simplified dispersion relation
of gravity waves by neglecting a wind shear (e.g., Nappo, 2002), such as
$$m^2 \approx \frac{N^2}{c_i^2} - \frac{1}{4H_s^2} - k^2, \qquad (2)$$
where N is the Brunt-Väisälä frequency, $c_i$ the is intrinsic phase speed of gravity wave, and
$H_s$ is the scale height. The intrinsic phase speed, $c_i$, can be expressed as c-u, where c is the
wave phase speed and u is the background wind speed in the wave propagating direction. The
Brunt-Väisälä frequency is given as
$$N^2 = \frac{g}{T}\left(\frac{dT}{dz} + \frac{g}{C_p}\right), \qquad (3)$$
where $g$ is the gravity, 9.55 $m/s^2$, T is the atmospheric temperature, $C_p$ is a specific heat
capacity at constant pressure, adopted as 1005 $J/(K \cdot Kg)$ for a dry air (Brasseur and Solomon,
2005). $H_s$ is given with $RT/g$, where R is the gas constant of dry air, 287 J/kg/K.
The intrinsic phase speeds of waves were computed by utilizing the wind at 96 km
simultaneously measured by the Gyeryong meteor radar. The intrinsic period is calculated by
$\lambda_h/c_i$, where $\lambda_h$ is the observed horizontal wavelength. The IQR of intrinsic phase speed of
gravity waves in spring is spanned from 15.7-67.3 m/s with a median value of 40.5 m/s, and
the IQR of intrinsic period is from 6.3- 21.8 min with a median of 11.5 min. In summer the
corresponding IQR values are 28.2-64.6 m/s with a median of 48.3 m/s, and 6.4 – 21.6 min
with a median of 11.5 min; in winter, the IQR values are 10.1 – 65.9 m/s with a median of 32.9
m/s, and 7.9 – 19.9 min with a median of 11.7 min. It is noted that the intrinsic speeds for spring
and summer are larger than the observed ones (see Figure 5), implying that the majority of
waves occurred in the opposite direction to the background wind. The intrinsic speed has been



merely shifted to the larger observed by the Doppler effect. The results of intrinsic parameters
exist in the typical values of gravity wave parameters such as intrinsic phase speeds of 30-100
m/s and intrinsic periods from 5-50 min (Taylor et al., 1997; Swenson et al., 2000; Hecht et al.,
2001; Ejiri et al., 2003).

6. Characteristics of vertical propagation

242        The nature of vertical propagation can be evaluated by the vertical wave number squared,

$m^2$ (Isler et al., 1997). If $m^2$ is greater than zero in the airglow-observed MLT region, the
gravity wave is in freely propagating mode. If $m^2$ is less than zero, the wave is vertically
evanescent, which indicates the wave motion only in the horizontal propagation. If the freely
propagating region is bounded by evanescent regions below and above, the wave is in ducting
mode. If it is bounded by one side evanescent region below or above, it is in partial ducting.

248        Based on the $m^2$ profile in 90-100 km centered at 96 km, the nature of vertical propagation

can be classified for seasons, as summarized in Table 2. Freely propagating waves take up a
maximum of 82% in summer and a minimum of 65% in spring. Ducted waves were 7% and
4% in summer and winter, respectively. Partial ducting takes up 28% and 20% for spring and
winter, respectively. Evanescent waves (7%) are observed only in spring. The small percentage
of evanescent waves may imply that the majority of the observed waves is not locally originated
from, at least the altitude range of 90 – 100 km. The freely propagating waves show vertical
wavelengths with a median value of 7.7 km and IQR ranged from 5.1-10.9 km. It should be
noted that the temperature profile used in the computation of the Brunt-Väisälä frequency is
the climatological one, not the real-time temperatures, which may result in the vertical
wavelengths smaller than the airglow layer thickness.
The wave ducting can be primarily caused by the background wind, so called Doppler ducting,
or primarily by a variation of Brunt-Väisälä frequency, so called thermal ducting. Since we use
the climatologic temperature profile, we cannot identify the thermal ducting that requires real-
time temperature measurements. On the other hand, Doppler ducting can be found rather
confidently because we use the simultaneously measured wind profile. Doppler ducting is
favorable when the wind profile has a local maximum against the wave propagation (e.g.,
Chimonas and Hines, 1986; Isler, 1997; and Nappo, 2002).
Examples of vertical propagation nature appraised by $m^2$ are shown in Figures 6a-c, where
the left panel presents the MR wind profile projected on the wave propagating direction, in
which the negative means that the wind blows opposite to the wave propagation direction, and
the right panel displays the $m^2$ profile.
In Figure 6a the gravity wave at a phase speed of c = 48.7 m/s was propagating northeastward
($\varphi$=52°) against the background wind at 90-100 km altitudes and the values of $m^2$ in the 90-
100 km region are all positive, indicating the freely propagating nature.
Figure 6b shows an example of a Doppler ducted wave. Here the freely propagating region ($m^2$
> 0) at 90-98 km is encompassed with the negative values of $m^2$ above 98 km and below 90
km. The winds opposing the gravity wave propagation becomes large above 98 km and lower
90 km. Therefore, the wave can be trapped around 96 km vertically, but still propagate to the
horizontal direction. Nielsen et al. (2012) noted that when jets occurred above and below the
altitude region of freely propagating ($m^2 > 0$), the wave can be bounded by evanescent regions
($m^2 < 0$), causing Doppler ducting. Suzuki et al. (2013) observed an evidence of Doppler
ducting under the large opposing winds: a northward propagating wave at a phase speed of 48
m/s lasting for ~5 hrs (11-17 UT) went through a strong southward wind, stretching over
16°x16° in latitude and longitude. For the ducted waves, it may be difficult to trace back the
source of the waves.
Figure 6c presents an example of an evanescent wave, based on negative values of $m^2$ in
the altitude range of 90- 97 km. The background wind is too fast in the opposing direction of
the wave, prohibiting the vertical propagation. The evanescent waves may be generated in situ
at the airglow layer, probably as secondary waves, not propagated from the lower atmosphere.
The evanescent waves were very rare (less than 2%) in our analysis of the BHO images. The
majority of observed waves are found to be freely propagating, and thus can be attributed to
wave sources in the lower atmosphere.



## 7. Summary and conclusions

This study investigated the characteristics of horizontal and vertical propagation of atmospheric gravity waves observed at Mt. Bohyun observatory (BHO, 36.2°N, 128.9°E) for the period of 2017 - 2019. The data used are all-sky images of the OI 557.7 nm airglow layer (~96 km). Wind data in the 80 -100 km altitude range measured by a meteor radar at a nearly co-located site were utilized to derive intrinsic wave parameters and their vertical propagation nature.

The results of our analysis can be summarized as follows:

1. The total of 150 wave events were identified in the images of 144 clear nights. The interquartile ranges (IQR) of wavelength, observed phase speed, and observed periods of the identified waves are 20.5 - 35.5 km (with a median value of 27.8 km), 27.4 - 45.0 m/s (with a median value of 36.3 m/s) and 10.8 -13.7 min (with 11.7 min median value), respectively.

2. The observed waves propagate predominantly northeastward and northward in spring and summer, respectively. In winter the majority of waves propagate southward but the significant portion of waves northward. The seasonal preferential directions as in our observation have been reported by previous studies in east Asia, and interpreted as the consequence of the critical level filtering effect due to the prevailing wind in the lower atmosphere. The observed northwestward waves in winter may be caused by nullified filtering effect due to small background wind field, secondary waves or waves generated in the upper mesosphere.

3. Intrinsic phase speeds and periods of the waves were also derived by using the wind data simultaneously observed by a meteor radar. It is noted that the intrinsic speeds for spring and summer are larger than the observed ones because the majority of waves propagate in the opposite direction to the background wind.

4. The nature of vertical propagation was evaluated in each season. The freely propagating waves take up a maximum of 82% in summer and a minimum of 65% in spring. Ducted waves were 7% and 4% in summer and winter, respectively. Evanescent waves were 7% only in spring. The majority of observed waves are found to be freely propagating, and





thus can be attributed to wave sources in the lower atmosphere.
In conclusion, we find that both horizontal and vertical propagation characteristics of the
observed waves at the OI 557.7 nm airglow layer are consistent with the notion that the majority
of waves originated from the lower atmosphere and experienced the filtering effect by the
prevailing winds in the intermediate atmosphere.

**Data availability**. We referred free reanalysis wind data from
https://gmao.gsfc.nasa.gov/reanalysis/MERRA-2/dataaccess/, last access: August 10, 2020,
for the mean wind field at altitudes of 0-80 km. We also used free model temperature data
from https://map.nrl.navy.mil/map/pub/nrl/NRLMSIS/NRLMSIS2.0/, last access August 20,
2021 as an element in making our figures and table.

**Supplement**. Not applicable.

**Author Contributions.** Y. H. Kim and Y.-S. Lee conceived of the presented idea and the design
of the study. J.-Y. Hwang and Y.-S. Lee manually gathered the data used. J.-Y. Hwang and Y.-
S. Lee programmed for data analysis. The data analysis and interpretation of the results were
done by Y. H. Kim and Y.-S. Lee. This paper was drafted and edited by Y.-S Lee and J.Y-.
Hwang, and critically reviewed by T.-Y. Yang, H. Kam, Y.-S. Kwak and Y. H. Kim for content.
Y.-S. Kwak and T.-Y. Yang took responsibility for overseeing the project. All authors have read
and agreed to the published version of the manuscript.

**Competing interests**. The contact author has declared that neither they nor their co-authors
have any competing interests.

**Disclaimer**. Publisher's note: Copernicus Publications remains neutral with regard to
jurisdictional claims in published maps and institutional affiliations.



**Acknowledgements.**
This research was supported by basic research funding from the Korea Astronomy and Space
Science Institute (KASI) (KASI2021185005). We would like to acknowledge the Geospace
Science & Technology Branch of U.S. Naval Research Laboratory (NRL) for providing the
MSIS2.0 model data (https://map.nrl.navy.mil/map/pub/nrl/NRLMSIS/NRLMSIS2.0/), and
NASA's Global Modeling and Assimilation Office for providing MERRA-2 atmospheric
reanalysis model. We would like to thank the anonymous reviewers for the critical reviews that
helped to improve this paper.

**Review statement**. This paper was edited by an editor (?) and reviewed by two anonymous
referees.














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





Figures
Figure 1 (a) an all-sky image with the OI 557.7 nm filter and (b) an image after star removal and
coordinate transformation. The image was observed at 15:32:33 UT on May 26,2017.
Figure 2. (a) A time-difference image (TD image) obtained by taking a subtraction between two
successive images, (b) an image after large-scale modulation removed from (a) by applying 2-D
bandpass filtering.
Figure 3. The parameters of the observed waves in the OI 557.7 airglow layer from 2017-2019, (a)
wavelength, (b) phase velocity, (c) period, and (d) propagation direction. Colors of blue, green, and
orange correspond to each year of 2017-1019, respectively.
Figures 4. Propagation vectors (left) and the occurrences (right) of observed waves in the OI airglow
over the three years from 2017 to 2019. (a) Spring, (b) Summer, (c) Fall and (d) Winter. The number on
the arc lines indicate (left) the phase velocity and (right) occurrences in each radial direction. Wave
propagation directions are divided into eight regions by a clockwise azimuth angle of 45° from -22.5°
to 315°, corresponding to the north (N), northeast (NE), east (E), southeast (SE), etc. In fall, both
equipment problem and poor weather resulted in particularly the small number of observations
comparing to other seasons.
Figure 5. Seasonal distributions of observed (yellow) and intrinsic (green) wave parameters. Each row
represents (a) Spring, (b) Summer, (c) Fall, and (d) Winter. Observed gravity waves are in total 150
events from April, 2017 to December, 2019, while intrinsic wave parameters were derived for 111
events when the wind data were available from the nearly co-located meter radar.
Figure 6. Examples of vertical propagation characteristics evaluated by vertical wave number squared,
$m^2$, and the relation with horizontal wind. (left) the background wind in the direction of the gravity
wave propagation and (right) the profile of $m^2$. (a) freely propagating, (b) Doppler ducted as
encompassed by negative $m^2$, (c) evanescent based on negative $m^2$ at 90-97 km. Each title noted with
the applied gravity wave occurring time, date and season. In addition, c and $\varphi$ indicate the phase speed
and azimuth angle of the horizontal propagation, respectively.














Figures

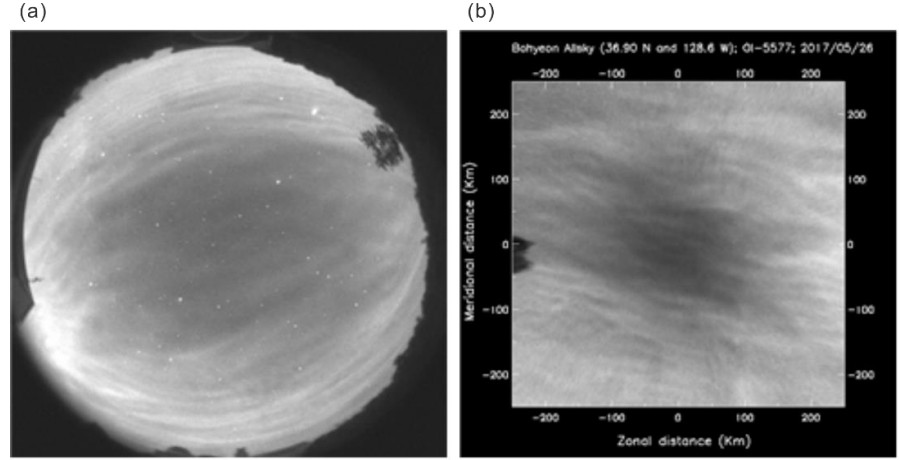


Figure 1.

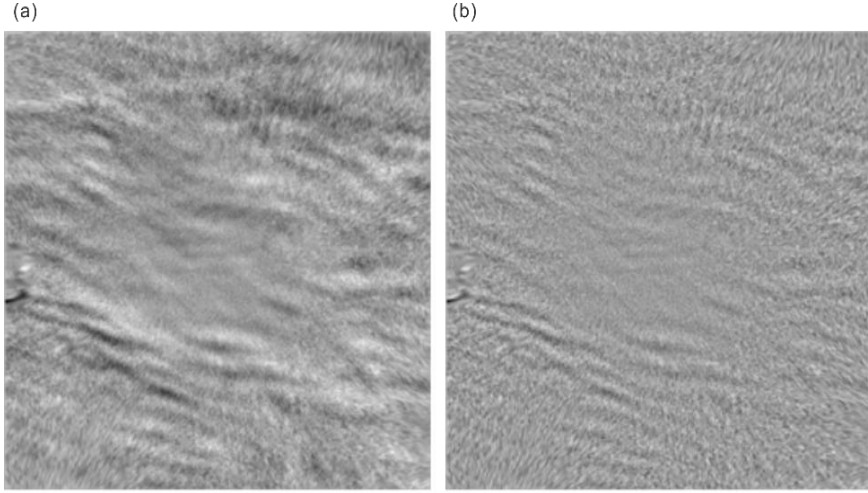


Figure 2.





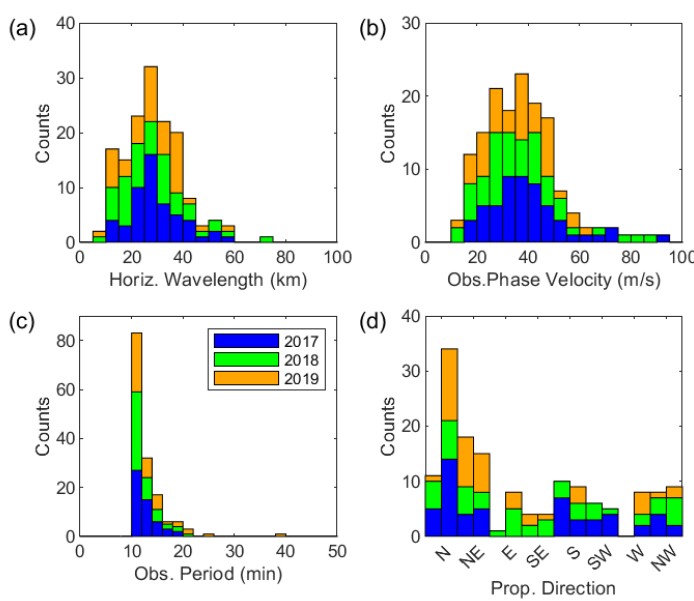


Figure 3.



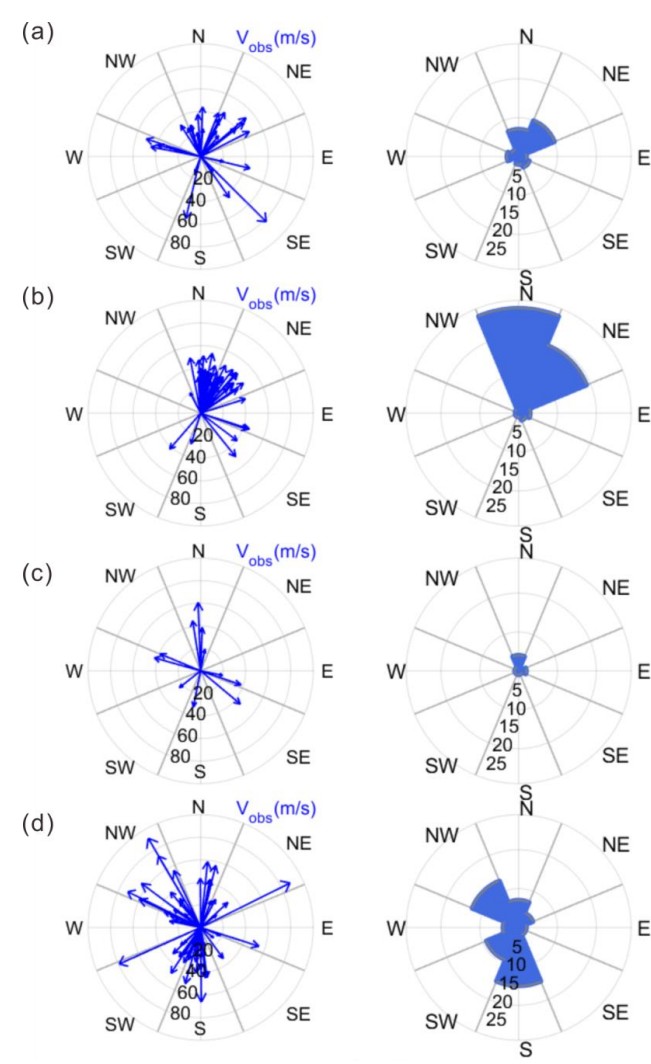


Figures 4.



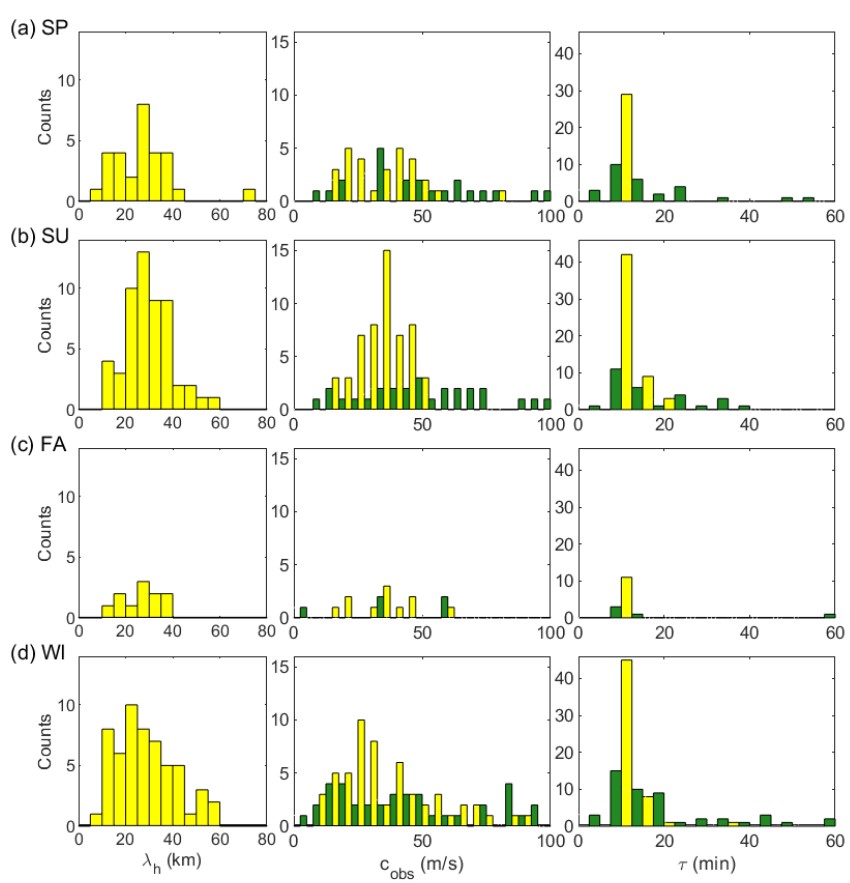


Figure 5.

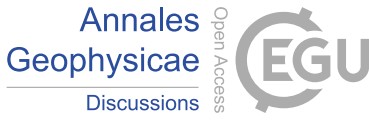

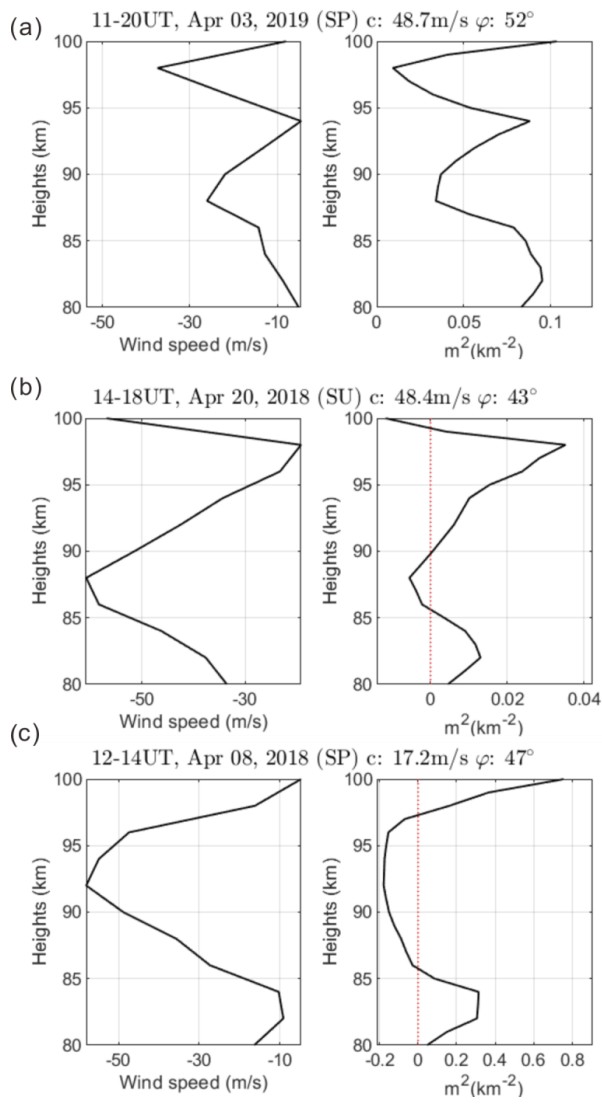


Figure 6.







Table 1. Seasonal median values and interquartile ranges (IQR) of wave parameters (observed
horizontal wavelength ($\lambda_{obs}$), observed phase speed ($c_{obs}$) and observed period ($\tau_{obs}$) observed at Mt.
Bohyun for 2017-2019.

| Seasons | Parameters | $\lambda_{obs}$ (km) | $c_{obs}$ (m/s) | $\tau_{obs}$ (min) |
|---------|--------|-------------|-------------|--------------|
| Spring | Median | 26.2 | 38.0 | 11.8 |
| | IQR | 18.0-31.4 | 24.8- 45.2 | 10.9- 13.6 |
| Summer | Median | 29.0 | 37.1 | 12.5 |
| | IQR | 23.7- 36.1 | 30.5- 42.1 | 11.3- 14.4 |
| Fall | Median | 25.7 | 38.7 | 11.7 |
| | IQR | 18.1-34.4 | 24.2- 45.1 | 10.7- 12.6 |
| Winter | Median | 27.5 | 32.7 | 11.5 |
| | IQR | 19.4-35.8 | 25.1- 46.7 | 10.8- 14.2 |


Table 2. Vertical propagation nature of gravity wave at Mt. Bohyun for 2017-2019.

| | Spring (%) | Summer (%) | Fall (%) | Winter (%) |
|---|---|---|---|---|
| Freely Propagating | 65 | 82 | 60 | 76 |
| Ducting | 0 | 7 | 20 | 4 |
| Partial Ducting | 28 | 11 | 20 | 20 |
| Evanescent | 7 | 0 | 0 | 0 |
| Total (no. events) | 29 | 28 | 5 | 49 |

