# Peer review of "Propagating characteristics of mesospheric gravity waves observed by an OI 557.7 nm airglow all-sky camera at Mt. Bohyun (36.2°N, 128.9°E)"

_Annales Geophysicae, 2021_

## Community Comment (CC1)

Response Letter

Thanks for the referee_#1's valuable comments.

This letter is to respond to the comments given by referee_#1.

The referee's comment is noted with C#X, and the response is given with the corresponding R#X.

C#1. The authors used "observed wave parameters "referring to the neutral wind Doppler

shifted values. I think it is better to refer them as "apparent".

R#1: Thanks for the indication. However, it needs to be reminded that gravity wave parameters such as observed phase speed, horizontal wavelength and period are commonly used in the research field of atmospheric gravity waves. Terminology of "apparent" is reasonable and applicable for the observed phase speed and period, but apparent wavelength is not proper, because wavelength is a solid measured value. Anyhow, the 'apparent' is supplementally noted for the observed phase speed and period in the context, as in line no. 133.

C#2: L182 "… Korean peninsula can propagate any directions …'. Change to '… in any

directions …'

R#2: It is corrected as indicated as in line no. 182.

C#3: L252 "The small percentage of evanescent waves may imply that the majority of the

observed waves is not locally originated from, at least the altitude range of $90 - 100$

km." Probably should just say that majority of the waves is free propagating wave.

R: It is corrected as using the recommended phrase as in line no. 253. Thanks for pointing out.

C#4: L286 "The evanescent waves may be generated in situ at the airglow layer, probably as

secondary waves, not propagated from the lower atmosphere. The evanescent waves

were very rare (less than 2%) in our analysis of the BHO images. "

- It will be good to show some examples of all sky images of the evanescent waves.

R#41: For the case of evanescent wave, gravity wave is almost not able to be distinguished with bare eyes from the all sky images probably due to low intensity of airglow. The wave outline is emerged as processed with using an image signal processing algorithm.

- As to the sources of the wave, do you have any references?

R#42: Simkhada, D.B., et al. (2009), Analysis and modeling of ducted and evanescent gravity waves observed in the Hawaiian airglow. Ann. Geophys. 27, 3213-3224.

Nielsen, K. et al (2012), On the nature of short-period mesospheric gravity wave propagation over Halley, Antarctica, J. Geophysical Res. 117, D05124.

Referring to Simkhada et al. (2009) the following is added in line 286-294: "Observed horizontal period is 11.8 min, while the intrinsic period ($\tau_i$) is 3.2 min. The intrinsic frequency ($\omega_i = 2\pi/\tau$) is greater than Brunt- Väisälä frequency which is typical for the atmospheric layer of evanescent wave occurring. The evanescent waves may be formed by the wave upward propagated from the

lower atmosphere, or by the secondary wave generated in situ at the airglow layer (e.g., Simkhada et al., 2009). Simkhada et al. (2009) presented a numerical result that evanescence occurs at 75-97 km from the wave forced below by the tropospheric source. It was diagnosed that the evanescent wave might be caused by encountering the opposing strong background wind field, generating a few m/s vertical wind to cause the perturbation at airglow layer."

Both papers presented all-sky images taken for clear evanescent gravity waves.

- Do you see any sign of the instability in the mesospheric region?

R#43: Airglow is too dim to determine atmospheric instability condition.

C#5: Did you do any binning of the CCD images?

R#5: All-sky images were obtained with 4×4 binning to increase signal to noise ratios.

The sentence is added in Line 94-95.

---

## Author Comment (AC1)

Response Letter

This letter is to respond to the comments given by referee #2.

The authors give thanks to referee #2 for the priceless comments leading to the improvement of the paper.

The referee's comments are noted with C#X, and the response is given in blue color, noted with the corresponding R#X.

Major comments:

C#1. The authors used NRLMSISE-00 temperature data, which is not useful for background analyses of individual cases. The author did not mention the role of thermal ducting due to the presence of mesospheric temperature inversion which plays a vital role in the ducting of GWs. Therefore, the statistical observations of free propagating, ducting, or evanescent waves are biased.

C#2. Model temperature cannot record any of these events. The authors should use SABER temperature data for this purpose.

R#1,2: Following the reviewer's comment, we reanalyze the vertical propagation characteristics by using temperature data measured by TIMED/SABER for Equation 2. We updated the results related to $m^2$. Total events of ducting and partial ducting are 9 for each, as shown in Table 3. Among those events Doppler vs. thermal ducting cases occurred with a ratio of 5:4. The advantage of using SABER temperature is to be able to specify either thermal or Doppler ducting, although the number of wave events is reduced down to 95 (from 111 previously) due to the limitation of the SABER coverage. The majority of vertical propagation is still in the freely propagating mode.

C#3. The seasonal results of propagation direction are similar to previously reported from the mid-latitude location. However, a serious question arises from the low number of data set for statistical analysis. On average, the observation day is only 13% of the total days of 3 years which is very small for statistical analyses. Can the authors justify this?

R#3: To some degree we agree with the reviewer's comment that the percentage of observation day is rather small for the seasonal statistical analysis, especially for fall season that we have already mentioned in the text. However, the total number of wave events we found is 150, more than the number for the 10% statistical uncertainty. Given that the optical observation is highly dependent on weather conditions, other studies (Espy et al. (2004, JGR) and Yang et al. (2010, JGR)) with all-sky camera have published the results with this level of number statistics.

C#4. In previous studies, the neutral instability has been attributed to the in-situ generations of ripples structure (<10-15 km). Figures 5a & 5d show the wavelength data in the range of 10 km. How are they sure that those small-scale waves are not ripples? In this regard, the authors are suggested to calculate the Richardson number to analyze dynamic instabilities.

R#4: Following the reviewer's comment, we calculated the Richardson number and found that only one event is satisfied with a shearing condition of Ri<0.25 in the direction of wave propagation. It is rather difficult to evaluate the instability from the MR measured wind data that were derived due to coarse temporal and spatial resolution (1 hr and 2 km altitude and few tens km horizontal bin resolution). Thus, we cannot claim any dynamical instability based on the Richardson number from the MR wind data.

C#5. Line 170-171: "…. the spring/summer the westward wind is dominant in the middle 171 atmosphere, whereas in the fall/winter the eastward wind is dominant …". The observation is during nighttime. Is the seasonal wind pattern mentioned above during nighttime? Authors are suggested to provide mean wind map or profiles of MERRA-2 over Mt. Bohyun observatory season-wise to support the conclusions in lines 321-324.

R#5: Following the reviewer's comment, we added the seasonal variation of nighttime winds for 2017-2019 from MERRA-2, as in Figure 6.

C#6. Why did the authors use interquartile range (IQR) instead of mean and standard deviation?

R#6: In the real situation, there are some cases not well fitted to the Gaussian distribution because the range of parameter distribution is so wide. For this case, IQR results are often used. We calculated the mean and standard deviation for the analysis result. However, intrinsic parameters frequently have large standard deviations, even greater than the mean value. In that case it is not meaningful to have the mean value. Thus, we prefer to use IQR with the median value, instead the mean and standard deviation. For more information, we provide additionally the mean and standard deviation in Table 1.

C#7. Is it possible to detect the nature of GWs (ducted, evanescent, or free propagating waves) based on airglow imager alone?

R#7: No. The airglow image provides only horizontal information. The vertical wave number can be estimated from the gravity wave relation (Eq. 2) with background wind and temperature information.

Minor comments:

Line 17: Please mention the bandwidth of the 557.7 nm filter.

R: The OI 557.7 nm filter has a central wavelength of 557.81 nm with a full-width half maximum of 1.53 nm as added in Line 96-97.

Line 217: The authors did not include the first and second derivative terms in the GWs.

R: The wind profile obtained from meteor radar is derived from the collected echoes for 1 hour with a 2 km height resolution. The wind data are interpolated to produce the wind profile of a 1 km resolution for the use of this study. In case that the first derivative and second derivative terms were added in the $m^2$ formula, the result would increase uncertainties seriously. Thus, we have to use the simplified version without those terms, as in Eq. (2).

Line 225: Please correct the scale height (Hs) and recalculate those profiles.

R: The scale height has been updated with using SABER temperature.

Line 252: What is the reason behind the observation of evanescent waves during spring

only?

R: We do not think that evanescent waves are particularly common or unique in spring, as shown in our new results in Table 2. Using the SABER temperature information, we found the evanescent waves in summer and winter, too.

Figure 6: The X-axis limits of all subplots in the left (wind speed) should be same and also

applicable to right subplots ($m^2$). It will be easy to compare.

R: The wind velocity is to be compared with the phase velocity of gravity wave, but not with other wind velocities. The $m^2$ is also the matter of change of negative and positive regions through the altitude. If the magnitudes are set with the same maximum from Figure 7a-d, the altitudinal variation would not be distinguishable to the reader.

---

## Author Comment (AC2)

Response Letter

Thanks for the referee_1's comment.

The author respond to the comments from referee_1.

Referee's comment is noted with C#X, and the response is given with the corresponding R#X.

C#1. The authors used "observed wave parameters " referring to the neutral wind Doppler

shifted values. I think it is better to refer them as "apparent".

R#1: gravity wave parameters such as observed phase speed, horizontal wavelength and period are commonly used in the society of atmospheric gravity waves. Apparent phase speed and apparent period are substituted with apparent ones, but apparent wavelength is not proper, because wavelength is a solid value. Anyhow, the 'apparent' is supplementally noted for the observed phase speed and period in the context.

C#2: L182 "… Korean peninsula can propagate any directions …'. Change to '… in any

directions …'

R#2: It is corrected as indicated.

C#3: L252 "The small percentage of evanescent waves may imply that the majority of the

observed waves is not locally originated from, at least the altitude range of 90 – 100

km." Probably should just say that majority of the waves is free propagating wave.

R#3: It is corrected as using the recommended phrase. Thanks for pointing out.

C#4: L286 "The evanescent waves may be generated in situ at the airglow layer, probably as

secondary waves, not propagated from the lower atmosphere. The evanescent waves

were very rare (less than 2%) in our analysis of the BHO images. "

- It will be good to show some examples of all sky images of the evanescent waves.

   For the case of evanescent wave, gravity wave is almost not able to be distinguished with bare eyes from the all sky images probably due to low intensity of airglow. The wave outline is emerged as processed with image signal processing.

- As to the sources of the wave, do you have any references?

   Simkhada, D.B., et al. (2009), Analysis and modeling of ducted and evanescent gravity waves observed in the Hawaiian airglow. Ann. Geophys. 27, 3213-3224.

   Nielsen, K. et al (2012), On the nature of short-period mesospheric gravity wave propagation over Halley, Antarctica, J. Geophysical Res. 117, D05124.

   The two papers present the raw image of evanescent wave. Simkhada et al. (2009) presented a result of modeling for evanescent wave. The evanescent wave is produced in the mesosphere at 75-95 km altitudes from the wave, which was forced in the troposphere and encountered the opposing strong background wind field while the upward propagation. It is added in lines 286-289. And The preexisted sentence is modified to "The evanescent waves may be formed by the

wave generated in the lower atmosphere, or by the secondary wave generated in situ at the airglow layer.".

Both papers presented all-sky images taken for clear evanescent gravity waves.

- Do you see any sign of the instability in the mesospheric region?

C#5: Did you do any binning of the CCD images?

All-sky images were obtained with 4×4 binning to increase signal to noise ratios.

The sentence is added in Line 94-95.

---

## Author Comment (AC3)

Response Letter 1

Thanks for the referee_#1's valuable comments.

This letter is to respond to the comments given by referee_#1.

The referee's comment is noted with C#X, and the response is given with the corresponding R#X.

C#1. The authors used "observed wave parameters "referring to the neutral wind Doppler shifted values. I think it is better to refer them as "apparent".

R#1: Thanks for the indication. The terminology of "apparent" seems reasonable and applicable for the observed phase speed and period, but apparent wavelength is not proper, because wavelength is a solid measured value. Gravity wave parameters such as observed phase speed, horizontal wavelength and period are commonly used by nomenclature in the research field of atmospheric gravity waves. Anyhow, the 'apparent' is supplementally noted for the observed phase speed and period in the context, as in line no. 44-145

"Along with the time difference and the wavelength information, the phase difference allows to determine the observed (apparent) phase speed of the dominant wave."

.

C#2: L182 "… Korean peninsula can propagate any directions …'. Change to '… in any directions …'

R#2: It is corrected as indicated as in line no. 194

"The typhoon-generated gravity waves in the south of the Korean peninsula can propagate in any directions,…"

C#3: L252 "The small percentage of evanescent waves may imply that the majority of the observed waves is not locally originated from, at least the altitude range of 90 – 100 km." Probably should just say that majority of the waves is free propagating wave.

R: Thanks for pointing out. It is corrected as using the recommended phrase as in line no. 253.

"The small percentage of evanescent waves may imply that the majority of the observed waves is freely propagating wave."

C#4: L286 "The evanescent waves may be generated in situ at the airglow layer, probably as secondary waves, not propagated from the lower atmosphere. The evanescent waves were very rare (less than 2%) in our analysis of the BHO images. "

- It will be good to show some examples of all sky images of the evanescent waves.

R#41: For the case of evanescent wave, gravity wave is almost not able to be distinguished with bare eyes from the all sky images probably due to low intensity of airglow. The wave outline is emerged as processed with using an image signal processing algorithm.

- As to the sources of the wave, do you have any references?

R#42: Simkhada, D.B., et al. (2009), Analysis and modeling of ducted and evanescent gravity waves observed in the Hawaiian airglow. Ann. Geophys. 27, 3213-3224.

Nielsen, K. et al (2012), On the nature of short-period mesospheric gravity wave propagation over Halley, Antarctica, J. Geophysical Res. 117, D05124.

Referring to Simkhada et al. (2009) the following is added in line 286-294: "Observed horizontal period is 11.8 min, while the intrinsic period ($\tau_i$) is 3.2 min. The intrinsic frequency ($\omega_i = 2\pi/\tau$) is greater than Brunt- Väisälä frequency which is typical for the atmospheric layer of evanescent wave occurring. The evanescent waves may be formed by the wave upward propagated from the lower atmosphere, or by the secondary wave generated in situ at the airglow layer (e.g., Simkhada et al., 2009). Simkhada et al. (2009) presented a numerical result that evanescence occurs at 75-97 km from the wave forced below by the tropospheric source. It was diagnosed that the evanescent wave might be caused by encountering the opposing strong background wind field, generating a few m/s vertical wind to cause the perturbation at airglow layer."

Both papers presented all-sky images taken for clear evanescent gravity waves.

- Do you see any sign of the instability in the mesospheric region?

R#43: Airglow is too dim to determine atmospheric instability condition.

C#5: Did you do any binning of the CCD images?

R#5: "All-sky images were obtained with 4×4 binning to increase signal to noise ratios."

The sentence is added in Line 94-95.

Response Letter 2

This letter is to respond to the comments given by referee #2.

The authors give thanks to referee #2 for the priceless comment to lead to an improvement of the paper.

The referee's comments are noted with C#X, and the response is given in blue color, noted with the corresponding R#X.

Major comments:

C#1. The authors used NRLMSISE-00 temperature data, which is not useful for background analyses of individual cases. The author did not mention the role of thermal ducting due to the presence of mesospheric temperature inversion which plays a vital role in the ducting of GWs. Therefore, the statistical observations of free propagating, ducting, or evanescent waves are biased.

C#2. Model temperature cannot record any of these events. The authors should use SABER temperature data for this purpose.

R#1,2: Following the reviewer's comment, we reanalyzed the vertical propagation characteristics by using temperature data measured by TIMED/SABER for Equation 2. We updated the results related to $m^2$. Total events of ducting and partial ducting are 9 for each, as shown in Table 3. Among those events Doppler vs. thermal ducting cases occurred with a ratio of 5:4. The advantage of using SABER temperature is to be able to specify either thermal or Doppler ducting, although the number of wave events is reduced down to 95 (from 111 previously) due to the limitation of the SABER coverage. The majority of vertical propagation is still in the freely propagating mode. According to the new results (Table 2 & 3 below), many of section 6 are updated.

Table 2. Vertical propagation nature of gravity waves at Mt. Bohyun for 2017-2019.

| | Spring (%) | Summer (%) | Fall (%) | Winter (%) |
|---|---|---|---|---|
| Freely Propagating | 60 | 89 | 0 | 69 |
| Ducting | 12 | 0 | 50 | 10 |
| Partial Ducting | 16 | 7 | 50 | 3 |
| Evanescent | 12 | 4 | 0 | 18 |
| Total (no. events) | 25 | 27 | 4 | 39 |

Table3. Classification of ducted and partially ducted waves into thermal or Doppler ducting.

| | Ducting event no. (9) | Partial ducting events no. (9) |
|---|---|---|
| Doppler ducting | 5 | 5 |
| Thermal ducting | 4 | 4 |

C#3. The seasonal results of propagation direction are similar to previously reported from the mid-latitude location. However, a serious question arises from the low number of data set for statistical analysis. On average, the observation day is only 13% of the total days of 3 years which is very small for statistical analyses. Can the authors justify this?

R#3: To some degree we agree with the reviewer's comment that the percentage of observation day is rather small for the seasonal statistical analysis, especially for fall season that we have already mentioned in the text. However, the total number of wave events we found is 150, more than the number for the 10% statistical uncertainty. Given that the optical observation is highly dependent on weather conditions, other studies (Espy et al. (2004, JGR) and Yang et al. (2010, JGR)) with all-sky camera have published the results with this level of number statistics.

C#4. In previous studies, the neutral instability has been attributed to the in-situ generations of ripples structure (<10-15 km). Figures 5a & 5d show the wavelength data in the range of 10 km. How are they sure that those small-scale waves are not ripples? In this regard, the authors are suggested to calculate the Richardson number to analyze dynamic instabilities.

R#4: Following the reviewer's comment, we calculated the Richardson number and found that only one event is satisfied with a shearing condition of Ri<0.25 in the direction of wave propagation. It is rather difficult to evaluate the instability from the MR measured wind data that were derived due to coarse temporal and spatial resolution (1 hr and 2 km altitude and few tens km horizontal bin resolution). Thus, we cannot claim any dynamical instability based on the Richardson number from the MR wind data.

C#5. Line 170-171: "…. the spring/summer the westward wind is dominant in the middle 171 atmosphere, whereas in the fall/winter the eastward wind is dominant …". The observation is during nighttime. Is the seasonal wind pattern mentioned above during nighttime? Authors are suggested to provide mean wind map or profiles of MERRA-2 over Mt. Bohyun observatory season-wise to support the conclusions in lines 321-324.

R#5: Following the reviewer's comment, we added the seasonal variation of nighttime winds for 2017-2019 from MERRA-2, as in Figure 6.

[Figure]

**Figure 6. Seasonal variations of nighttime (upper) zonal and (lower) meridional wind vectors in terms of day number (1-356) and 20-80 km (56-0.01 hPa) altitude from MERRA-2 reanalysis data for 2017-2019.**

C#6. Why did the authors use interquartile range (IQR) instead of mean and standard deviation?

R#6: In the real situation, there are some cases not well fitted to the Gaussian distribution because the range of parameter distribution is so wide. For this case, IQR results are often used. We calculated the mean and standard deviation for the analysis result. However, intrinsic parameters frequently have large standard deviations, even greater than the mean value. In that case it is not meaningful to have the mean value. Thus, we prefer to use IQR with the median value, instead the mean and standard deviation. For more information, we provide additionally the mean and standard deviation in Table 1.

C#7. Is it possible to detect the nature of GWs (ducted, evanescent, or free propagating waves) based on airglow imager alone?

R#7: No. The airglow image provides only horizontal information. The vertical wave number can be estimated from the gravity wave relation (Eq. 2) with background wind and temperature information.

Minor comments:

Line 17: Please mention the bandwidth of the 557.7 nm filter.

R: The OI 557.7 nm filter has a central wavelength of 557.81 nm with a full-width half maximum of 1.53 nm as added in Line 96-97.

Line 217: The authors did not include the first and second derivative terms in the GWs.

R: The wind profile obtained from meteor radar is derived from the collected echoes for 1 hour with a 2 km height resolution. The wind data are interpolated to produce the wind profile of a 1 km resolution for the use of this study. In case that the first derivative and second derivative terms were added in the $m^2$ formula, the result would increase uncertainties seriously. Thus, we have to use the simplified version without those terms, as in Eq. (2).

Line 225: Please correct the scale height (Hs) and recalculate those profiles.

R: The scale height has been updated with using SABER temperature.

Line 252: What is the reason behind the observation of evanescent waves during spring

only?

R: We do not think that evanescent waves are particularly common or unique in spring, as shown in our new results in Table 2. Using the SABER temperature information, we found the evanescent waves in summer and winter, too.

Figure 6: The X-axis limits of all subplots in the left (wind speed) should be same and also

applicable to right subplots ($m^2$). It will be easy to compare.

R: The wind velocity is to be compared with the phase velocity of gravity wave, but not with other wind velocities. The $m^2$ is also the matter of change of negative and positive regions through the altitude. If the magnitudes are set with the same maximum from Figure 7a-d (as shown below), the altitudinal variation would not be distinguishable to the reader.

[Figure]

Figure7 (caption): Examples of vertical propagation characteristics evaluated by vertical wave number squared, m$^2$, and the relation with temperature and horizontal wind. (left) the profile of m$^2$; (middle) Wind speed aligned to wave propagating direction and Temperature (red); (right) Brunt-Väisälä frequency squared (N$^2$, red) and intrinsic frequency squared ($\omega^2$). (a) freely propagating, (b) evanescent based on negative m$^2$ at 90-97 km, (c) Ducted as encompassed by negative m$^2$, and (d) Partially ducted by above m$^2$. Each title noted with the applied gravity wave occurring time, date and season. In addition, c and $\varphi$ indicate the apparent phase speed and azimuth angle of the horizontal propagation, respectively.

---

## Author Response (AR2)

**Response to Reviewers**

Dear Editor,

We appreciate the time and effort that you and the reviewers have dedicated to providing valuable comments. It owes to your valuable and insightful comments to have led our paper to possible improvements as the current version. The authors have carefully considered and tried our best to address the comments. Below we provide the point-by-point responses. All modifications in the manuscript have been highlighted in blue.

**Comments from Reviewer 1**

**Comment:** *I only suggest that the authors include some images of evanescent waves even if they may not be visible. I think other readers may be curious as well.*

**Response:** Thank you for this suggestion. It would have been considerable if the image taken at Bohyun were comparable to those of previous papers including

1. Nielsen, K., Taylor, M. J.,Hibbins, R.E., Jarvis, M.J. and Russell III, J.M.: On the nature of short-period mesospheric gravity wave propagation over Halley, Antarctica, J. Geophys. Res., 117, D05124, doi:10.1029/2011JD016261, 2012, and

2. Isler, J. R., Taylor, M.J., and Fritts, D.C.: Observational evidence of wave ducting and evanescence in the mesosphere, J. Geophys. Res., 102, 26,301–26,313, 1997.

The referring papers present vivid images of evanescent waves with carrying excellent outlines of the waves. However, it is unable to easily grasp with bare eyes any outlines of wave forms in the raw images, which were directly taken by camera at Bohyun. Instead, we revised with adding the citations at the points delivering the results of evanescent and ducted waves, as in lines 226-228

"The evanescent waves (e.g., Nielsen et al., 2012; Isler et al., 1997) may be formed by the wave upward propagating from the lower atmosphere…".

**Comment:** *The authors have incorporated all the suggestions within their limitations and the quality of the paper is improved. Therefore, I recommend this manuscript for publication*

**Answer:** Thanks for the considerable recommendation.

Minor Comments /Suggestions:

**Comment:** *1. Line -19-20: "The majority of observed waves are found to be freely propagating, and thus can be attributed to wave sources in the lower atmosphere"*

*Why can sources of freely propagating waves be attributed in the lower atmosphere? In my opinion, the filtering effect and preferential direction of GWs propagation in the MLT region should be attributed to identifying the source in the lower or upper atmosphere.*

**Response:** Thanks for the pointing this out. We agree with your suggestion, and we reflected your suggestion in the manuscript as in lines 19-20

"The majority of observed waves are found to be freely propagating, and thus can be attributed to wave sources in the lower or upper atmosphere."

**Comment:** *2. I am wondering why most of the waves are free propagating at Mt. Bohyun (36.2°N, 128.9°E). In contrast, a previous study by Isler et al., 1997 indicates a preponderance of duct or evanescent waves up to 70 % of recorded events at Hawaiian Island.*

*Reference:*

*Isler, J. R., Taylor, M. J., & Fritts, D. C. (1997). Observational evidence of wave ducting and evanescence in the mesosphere. Journal of Geophysical Research: Atmospheres, 102(D22), 26301-26313.*

**Response:** The observational results given by Isler et al. (1997) were from the images taken in a Fall season only from October 6-22, 1993. In addition, they used four different filters to estimate the vertical propagation of gravity wave, covering wider altitudinal range from $87\pm10$km (OH) to $96\pm10$ km OI (557.7 nm). In comparison to this, our observation was taken except for Fall due to instrumental and weather problems with using single filter of OI (557.7 nm). Besides, Korean Peninsular is located at a continental region in which geographical location and climate effect are different from those of Hawaii Island. Therefore, the characteristic results of vertical gravity wave propagation should be different between two sites.